# Efficient Federated Incomplete Multi-View Clustering

Suyuan Liu [1]  Hao Yu [1]  Hao Tan [1]  Ke Liang [1]  Siwei Wang [2]  Shengju Yu [1]  En Zhu [1]  Xinwang Liu [1]

## Abstract

Multi-view clustering (MVC) leverages complementary information from diverse data sources to enhance clustering performance. However, its practical deployment in distributed and privacy-sensitive scenarios remains challenging. Federated multi-view clustering (FMVC) has emerged as a potential solution, but existing approaches suffer from substantial limitations, including excessive communication overhead, insufficient privacy protection, and inadequate handling of missing views. To address these issues, we propose Efficient Federated Incomplete Multi-View Clustering (EFIMVC), a novel framework that introduces a localized optimization strategy to significantly reduce communication costs while ensuring theoretical convergence. EFIMVC employs both view-specific and shared anchor graphs as communication variables, thereby enhancing privacy by avoiding the transmission of sensitive embeddings. Moreover, EFIMVC seamlessly extends to scenarios with missing views, making it a practical and scalable solution for real-world applications. Extensive experiments on benchmark datasets demonstrate the superiority of EFIMVC in clustering accuracy, communication efficiency, and privacy preservation. Our code is publicly available at https://github.com/Tracesource/EFIMVC.

## 1. Introduction

Multi-view clustering (MVC) has garnered significant attention in the field of unsupervised learning, primarily due to the increasing availability of diverse data from various sources or perspectives (Cui et al., 2024a; Liang et al., 2023; Liu et al., 2021). Unlike traditional clustering methods, which typically operate on a single data representation, MVC integrates complementary and consensus information from multiple views, thereby significantly enhancing clustering robustness and accuracy (Yang et al., 2022; Wang et al., 2013; Pan & Kang, 2021; Wan et al., 2024a). Over the past decade, numerous MVC approaches have been developed, refining performance from different perspectives (Sun et al., 2024; Cui et al., 2024b; Wan et al., 2024b).

Despite these advancements, most existing MVC methods assume a centralized data storage paradigm. In practical applications, however, multi-view data are often distributed across multiple clients due to privacy concerns, regulatory constraints, or logistical limitations (Che et al., 2022; Huang et al., 2022). Direct data sharing among clients is typically infeasible, making traditional MVC approaches unsuitable for such scenarios. To bridge this gap, Federated Multi-View Clustering (FMVC) has been introduced, combining i (FL) with MVC to enable privacy-preserving, distributed clustering (Feng et al., 2024b; Chen et al., 2024). For instance, Feng et al. (Feng et al., 2024a) integrated matrix factorization and K-Means clustering within a federated learning framework, while Jiang et al. (Jiang et al., 2024) proposed contrastive learning-based local-global model alignment to handle data heterogeneity in FMVC.

However, existing FMVC methods still face several critical challenges: **1) High Communication Overhead.** All existing FMVC frameworks rely on iterative exchanges of client- and server-side variables within tightly coupled optimization processes. Each update step necessitates a communication round, making the communication cost proportional to the number of optimization iterations. Furthermore, some methods transmit large $n \times n$ similarity matrices, exacerbating bandwidth consumption. **2) Limited Privacy Protection.** Most FMVC approaches employ either embedded representations or sample-level similarity matrices as communication variables. While embeddings encapsulate feature information, similarity matrices reveal the intrinsic relationships between samples. Both contain substantial view-specific private information, posing privacy risks. **3) Ineffective Handling of Missing Views.** Real-world multiview data often suffer from missing modalities due to sensor failures, transmission errors, or incomplete data collection. Many existing FMVC methods assume complete multi-view data, rendering them ineffective in handling missing views.

[1]College of Computer Science and Technology, National University of Defence Technology, Changsha, China [2]Academy of Military Sciences, Beijing, China. Correspondence to: Xinwang Liu <xinwangliu@nudt.edu>.

*Proceedings of the 42nd International Conference on Machine Learning*, Vancouver, Canada. PMLR 267, 2025. Copyright 2025 by the author(s).

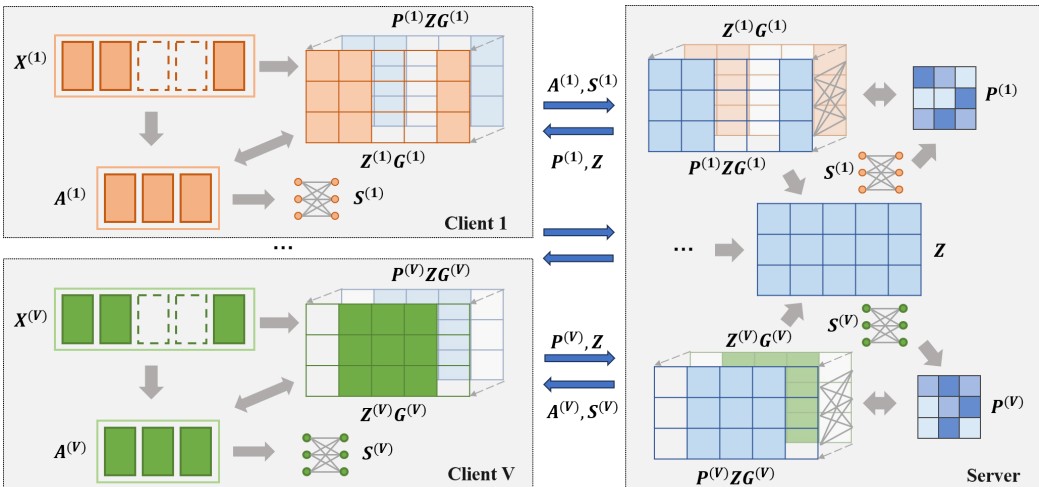

*Figure 1.* Overview of the proposed EFIMVC framework. EFIMVC follows an alternating local optimization strategy between clients and the server. Clients perform single-view anchor learning and anchor graph construction, guided by the global anchor graph from the server. The optimized anchor graph and anchor relationship matrix are then sent to the server, which aligns and fuses the received anchor graphs through local optimization. The updated global anchor graph and alignment matrices are transmitted back to clients, and this process continues until global convergence.

To tackle these challenges, we propose Efficient Federated Incomplete Multi-View Clustering (EFIMVC), a novel FMVC framework designed to enhance communication efficiency, privacy protection, and robustness to missing views. As illustrated in Figure 1, EFIMVC introduces a federated optimization strategy that confines variable updates to local environments, thereby reducing communication overhead while ensuring theoretical convergence. By leveraging view-specific and shared anchor graphs as communication variables, EFIMVC mitigates bandwidth demands and avoids the direct transmission of sensitive embeddings, enhancing privacy protection. Furthermore, we introduce a dual-anchor alignment mechanism based on structural and feature consistency to mitigate anchor misalignment during graph fusion. Finally, EFIMVC seamlessly extends to incomplete multi-view scenarios, offering a practical solution for real-world federated clustering tasks.

The key contributions of EFIMVC are summarized as follows:

- We propose a novel federated optimization framework that minimizes communication overhead by confining variable updates to local environments, significantly improving communication efficiency without sacrificing clustering performance.

- To enhance privacy, EFIMVC employs anchor graphs as communication variables, avoiding the transmission of sensitive embeddings or sample-level similarity matrices, thereby reducing privacy risks.

- We introduce a dual-anchor alignment mechanism to

address anchor misalignment during graph fusion, improving the robustness and reliability of clustering results.

- EFIMVC effectively handles missing views, ensuring robust clustering performance even in the presence of incomplete multi-view data.

## 2. Related Work

### 2.1. Federated Multi-view Clustering

Federated Multi-View Clustering has emerged as a promising approach for exploring the cluster structure of multi-view data distributed across multiple clients while preserving data privacy (Yan et al., 2024; Chen et al., 2023). Existing FMVC methods can be categorized based on the type of communication variables exchanged between clients and the server.

One common approach is to transmit embedded representations, where each client sends view-specific embeddings of size $n \times d$ (with $n$ as the sample size and $d$ as the feature dimension). Chen et al. (Chen et al., 2024) proposed a framework integrating local collaborative contrastive learning and global weighted aggregation to address challenges posed by heterogeneous hybrid-view scenarios. Similarly, Yan et al. (Yan et al., 2024) introduced a heterogeneous graph neural network and global pseudo-labeling strategy to mitigate view incompleteness and feature heterogeneity, demonstrating the effectiveness of embedding-based communication. However, this approach risks privacy leakage since embeddings may retain sensitive feature information.

Another strategy involves transmitting similarity matrices of size $n \times n$, encoding the structural relationships between samples. For instance, Wang et al. (Wang et al., 2020) proposed a federated multi-view spectral clustering method that leverages similarity matrices to enable distributed spectral clustering. While this approach helps retain local structure information, the transmission of full similarity matrices can lead to excessive communication costs, especially for large-scale datasets.

A third category of FMVC methods exchanges partition matrices, where each client transmits a clustering partition matrix of size $n \times k$ (with $k$ as the number of clusters). Feng et al. (Feng et al., 2024a) integrated matrix factorization and multi-view $k$-means to address the separation between feature extraction and clustering. Feng et al. (Feng et al., 2024b) further proposed a tensor factorization-based FMVC framework to enhance multi-view data integration and improve clustering stability. Partition matrix-based methods often strike a balance between privacy and communication efficiency, but they may still struggle with view inconsistencies and incomplete data.

Despite these advancements, existing FMVC methods suffer from high communication overhead, privacy risks due to the transmission of sensitive information, and poor handling of missing views. Addressing these challenges remains a key focus of federated multi-view clustering research.

### 2.2. Anchor-based Multi-view Clustering

To mitigate the computational complexity associated with large-scale multi-view clustering, anchor-based MVC methods have been proposed (Wang et al., 2023; Yan et al., 2022; Zhang et al., 2023b). These methods construct a compact set of representative anchors to approximate the data structure, significantly reducing the cost of similarity computations (Liu et al., 2022; Zhang et al., 2023a).

Traditional anchor-based MVC methods rely on sampling-based anchor selection strategies (Qiang et al., 2021; Shi et al., 2021; Yang et al., 2020). More recent approaches focus on learning anchor representations to improve clustering quality (Liu et al., 2024; Ma et al., 2024; Wu et al., 2023; Qin et al., 2024). By replacing pairwise sample similarities with anchor-sample relationships, these methods reduce both storage and computational overhead, making them particularly well-suited for scalable clustering applications.

Unlike traditional MVC approaches that establish relationships among all samples, anchor-based methods optimize clustering by focusing only on anchor relationships. This facilitates efficient multi-view fusion and clustering while maintaining high accuracy and robustness. In the context of federated learning, integrating anchor-based clustering with FMVC can further reduce communication overhead

and enhance privacy protection.

## 3. Methodology

### 3.1. Motivation

Federated Multi-View Clustering aims to alleviate the computational burden on individual clients by distributing the learning process across multiple nodes while preserving data privacy. Unlike traditional multi-view clustering approaches, FMVC introduces significant communication overhead, which becomes a critical bottleneck affecting its efficiency. The communication overhead is determined by two key factors: communication frequency, i.e., the number of communication rounds, and communication bandwidth, i.e., the amount of data transmitted per round. Existing FMVC methods generally transmit local variables immediately after each update, leading to a communication frequency proportional to the number of optimization iterations. Regarding communication bandwidth, most methods fall into three categories: transmission of embedded representations, similarity matrices, or clustering partition matrices. While embedded representations are more space-efficient than similarity matrices, they can directly expose local client data, posing a risk of privacy leakage. Furthermore, existing FMVC methods are primarily designed for complete multi-view data and struggle to handle scenarios with missing views. To address these limitations, we propose an Efficient Federated Incomplete Multi-View Clustering (EFIMVC) framework that effectively balances communication efficiency, privacy preservation, and robustness to missing views

### 3.2. Local Learning in Clients

To mitigate the high communication frequency in existing FMVC methods, we introduce a novel framework that decouples local optimization from global optimization. In our approach, both clients and the server maintain their respective optimization objectives. Instead of transmitting updates at every iteration, communication occurs only after local optimization converges to a stable state, reducing unnecessary transmissions. Additionally, to lower communication bandwidth, we design the transmitted variables as anchor graphs, where the view-specific encoding of these graphs safeguards client data privacy.

Specifically, given the data $\mathbf{X}^{(v)} \in \mathbb{R}^{d_v \times n}$ for the $v$-th view, the local objective function on the corresponding client is formulated as follows.

$$\min_{\mathbf{A}^{(v)}, \mathbf{Z}^{(v)}} \left\| \mathbf{X}^{(v)} \mathbf{G}^{(v)} - \mathbf{A}^{(v)} \mathbf{Z}^{(v)} \mathbf{G}^{(v)} \right\|_{\mathbf{F}}^2, \\ \text{s.t. } (\mathbf{Z}^{(v)})^\top \mathbf{1} = \mathbf{1}, \mathbf{Z}^{(v)} \geq 0, \tag{1}$$

where $\mathbf{A}^{(v)} \in \mathbb{R}^{d_v \times m}$ represents the anchor matrix, and

$\mathbf{Z}^{(v)} \in \mathbb{R}^{m \times n}$ is the incomplete anchor graph that encodes the relationships between anchors and the available sample points. The column-sum constraint ensures that each sample can be expressed as a linear combination of all anchors. Additionally, an index matrix $\mathbf{G}^{(v)} \in \{0,1\}^{n \times n_v}$ is introduced to indicate the positions of available samples in missing views (Wang et al., 2022). Importantly, Eq. (1) leverages only local single-view information.

To further guide the learning process with global information, we incorporate a federated alignment term that enforces consistency between local and global anchor graphs as follows,

$$
\begin{aligned}
\min_{\mathbf{A}^{(v)}, \mathbf{Z}^{(v)}} & \left\| \mathbf{X}^{(v)} \mathbf{G}^{(v)} - \mathbf{A}^{(v)} \mathbf{Z}^{(v)} \mathbf{G}^{(v)} \right\|_{\mathbf{F}}^2 \\
& + \lambda \left\| \mathbf{P}^{(v)} \mathbf{Z} \mathbf{G}^{(v)} - \mathbf{Z}^{(v)} \mathbf{G}^{(v)} \right\|_{\mathbf{F}}^2, \\
& \text{s.t. } (\mathbf{Z}^{(v)})^\top \mathbf{1} = \mathbf{1}, \mathbf{Z}^{(v)} \geq 0,
\end{aligned}
\tag{2}
$$

where $\mathbf{Z} \in \mathbb{R}^{m \times n}$ denotes the global anchor graph, and $\mathbf{P}^{(v)} \in \mathbb{R}^{m \times m}$ is an alignment matrix ensuring structural consistency across views. By enforcing dual alignment at both the anchor and sample levels with the global anchor graph $\mathbf{Z}$, the learning of the local anchor graph $\mathbf{Z}^{(v)}$ is effectively guided.

Once the local learning process is completed, each client computes an anchor similarity matrix $\mathbf{S}^{(v)} \in \mathbb{R}^{m \times m}$ based on the pairwise distances between anchor features. The element of $\mathbf{S}_{i,j}^{(v)}$ at the $i$-th row and $j$-th column is calculated as follows:

$$
\mathbf{S}_{i,j}^{(v)} = exp \left( - \frac{\left\| \mathbf{A}_{:,i}^{(v)} - \mathbf{A}_{:,j}^{(v)} \right\|_2^2}{2\sigma^2} \right).
\tag{3}
$$

Finally, the learned anchor graph $\mathbf{Z}^{(v)}$ and similarity matrix $\mathbf{S}^{(v)}$ are transmitted to the server for global fusion.

### 3.3. Global Fusion in Server

Upon receiving the anchor graphs from all clients, the server aggregates them to construct a unified global anchor graph. However, due to independent learning processes across different views, anchor misalignment naturally arises, necessitating an additional alignment mechanism. To address this issue, we introduce an alignment matrix $\mathbf{P}^{(v)}$ to ensure structural consistency between the global anchor graph and view-specific anchor graphs:

$$
\begin{aligned}
\min_{\mathbf{Z}, \mathbf{P}^{(v)}} & \sum_{v=1}^{V} \left\| \mathbf{P}^{(v)} \mathbf{Z} \mathbf{G}^{(v)} - \mathbf{Z}^{(v)} \mathbf{G}^{(v)} \right\|_{\mathbf{F}}^2, \\
& \text{s.t. } \mathbf{Z}^\top \mathbf{1} = \mathbf{1}, \mathbf{Z} \geq 0, (\mathbf{P}^{(v)})^\top \mathbf{P}^{(v)} = \mathbf{I},
\end{aligned}
\tag{4}
$$

where $\mathbf{G}^{(v)}$ extracts the columns corresponding to existing points in each view to participate in the anchor graph fusion.

By ensuring both row and column alignment, the accuracy of the fused information is guaranteed.

Additionally, inspired by manifold learning principles, we enforce structural consistency by incorporating anchor similarity constraints. Specifically, the similarity between anchors in each view should be preserved when aligning them with the global anchor graph. This ensures that similar anchors maintain consistent clustering structures across views. The local objective function in server is as follows,

$$
\begin{aligned}
\min_{\mathbf{Z}, \mathbf{P}^{(v)}} & \sum_{v=1}^{V} \left\| \mathbf{P}^{(v)} \mathbf{Z} \mathbf{G}^{(v)} - \mathbf{Z}^{(v)} \mathbf{G}^{(v)} \right\|_{\mathbf{F}}^2 \\
& + \beta \left\| (\mathbf{P}^{(v)} \mathbf{Z})_{i,:} - (\mathbf{P}^{(v)} \mathbf{Z})_{j,:} \right\|_2^2 \mathbf{S}_{i,j}^{(v)}, \\
& \text{s.t. } \mathbf{Z}^\top \mathbf{1} = \mathbf{1}, \mathbf{Z} \geq 0, (\mathbf{P}^{(v)})^\top \mathbf{P}^{(v)} = \mathbf{I}.
\end{aligned}
\tag{5}
$$

From the perspective of anchor alignment, the first term ensures row-level alignment by matching the structural similarity between the unified and view-specific anchor graphs, while the second term enforces structure-level alignment using anchor similarity.

### 3.4. Optimization

Unlike previous approaches that alternately optimize all variables on both the client and server within the same framework, our method ensures that variables at one end are transmitted only after being optimized to their local optimum. To achieve this, we propose two alternating optimization algorithms to solve Eq. (2) and Eq. (5). In each iteration, we optimize one variable while keeping the others fixed at their values from the previous iteration.

#### 3.4.1. Optimization in the $v$-th Clients

**Update $\mathbf{A}^{(v)}$:** Fixing $\mathbf{Z}^{(v)}$, Eq. (2) is transferred to the optimization problem related to $\mathbf{A}^{(v)}$:

$$
\max_{\mathbf{A}^{(v)}} \text{Tr} \left( \mathbf{A}^{(v)} \mathbf{C}^{(v)} (\mathbf{A}^{(v)})^\top - 2(\mathbf{A}^{(v)})^\top \mathbf{D}^{(v)} \right),
\tag{6}
$$

where $\mathbf{C}^{(v)} = \mathbf{Z}^{(v)} \mathbf{B}^{(v)} (\mathbf{Z}^{(v)})^\top$, $\mathbf{D}^{(v)} = \mathbf{X}^{(v)} \mathbf{B}^{(v)} (\mathbf{Z}^{(v)})^\top$, $\mathbf{B}^{(v)} = \mathbf{G}^{(v)} (\mathbf{G}^{(v)})^\top$. By taking the derivative of Eq. (6) with respect to $\mathbf{A}^{(v)}$ and setting it to zero, we get the closed-form solution for $\mathbf{A}^{(v)}$ as follows:

$$
\mathbf{A}^{(v)} = \mathbf{D}^{(v)} \mathbf{C}^{(v)^{-1}}.
\tag{7}
$$

**Update $\mathbf{Z}^{(v)}$:** Fixing $\mathbf{A}^{(v)}$, We get the optimization problem related to $\mathbf{Z}^{(v)}$ as follows:

$$
\begin{aligned}
\min_{\mathbf{Z}^{(v)}} \text{Tr} \Big( & (\mathbf{Z}^{(v)})^\top (\mathbf{A}^{(v)})^\top \mathbf{A}^{(v)} \mathbf{Z}^{(v)} \mathbf{B}^{(v)} + \lambda (\mathbf{Z}^{(v)})^\top \mathbf{Z}^{(v)} \mathbf{B}^{(v)} \\
& - 2((\mathbf{X}^{(v)})^\top \mathbf{A}^{(v)} + \lambda (\mathbf{P}^{(v)} \mathbf{Z})^\top) \mathbf{Z}^{(v)} \mathbf{B}^{(v)})^\top \Big), \\
& \text{s.t. } (\mathbf{Z}^{(v)})^\top \mathbf{1} = \mathbf{1}, \mathbf{Z}^{(v)} \geq 0.
\end{aligned}
\tag{8}
$$

**Algorithm 1** The proposed EFIMVC

**Input:** Incomplete multi-view datasets $\{\mathbf{X}^{(v)}\}_{v=1}^{V}$ and the corresponding index matrix $\{\mathbf{G}^{(v)}\}_{v=1}^{V}$ on $V$ local clients, the number of anchors $m$, the parameter $\lambda$, $\beta$, and the number of clusters $k$.

**Output:** Clustering results.

```
// Initialization
```
Initialize $\mathbf{A}^{(v)}$, $\mathbf{Z}^{(v)}$, $\mathbf{P}^{(v)}$ and $\mathbf{Z}$;
**while** *not converged* **do**
    `// Clients Optimization`
    **for** $v = 1 \rightarrow V$ **do**
        **while** *not converged* **do**
            Update $\mathbf{A}^{(v)}$ with Eq. (7);
            Update $\mathbf{Z}^{(v)}$ with Eq. (10);
        Calculate $\mathbf{S}^{(v)}$ according Eq. (3);
        Send $\mathbf{Z}^{(v)}$ and $\mathbf{S}^{(v)}$ to server;
    `// Server Optimization`
    **while** *not converged* **do**
        **for** $v = 1 \rightarrow V$ **do**
            Update $\mathbf{P}^{(v)}$ with Eq. (12);
        Update $\mathbf{Z}$ with Eq. (14);
    **for** $v = 1 \rightarrow V$ **do**
        Send $\mathbf{P}^{(v)}$ and $\mathbf{Z}$ to the $v$-th client.

```
// Final Results Generation
```
Perform $k$-means on the left singular vector of $\mathbf{Z}$.

*Table 1.* Datasets description.

| Dataset | $n$ | $V$ | $k$ | $d_v(v = 1, \ldots, V)$ |
|---|---|---|---|---|
| ProteinFold | 694 | 12 | 27 | 27/.../27 |
| WebKB | 1051 | 2 | 2 | 334/2949 |
| 100Leaves | 1600 | 3 | 100 | 64/64/64 |
| CCV | 6773 | 3 | 20 | 20/20/20 |
| YTF10 | 38654 | 4 | 10 | 512/576/640/944 |
| CIFAR10 | 50000 | 3 | 10 | 512/1024/2048 |
| MNIST | 60000 | 3 | 10 | 64/342/1024 |

To solve the optimization problem in Eq. (11), we reformulate it into the following equivalent form:

$$
\begin{aligned}
\max_{\mathbf{P}^{(v)}} &\operatorname{Tr}\big(\beta(\mathbf{P}^{(v)})^{\top}(\gamma_{max}\mathbf{I} - \mathbf{L}^{(v)})\mathbf{P}^{(v)}\mathbf{Z}\mathbf{Z}^{\top} \\
&+ (\mathbf{P}^{(v)})^{\top}\mathbf{Z}^{(v)}\mathbf{B}^{(v)}\mathbf{Z}^{\top}\big), \\
&\text{s.t. } (\mathbf{P}^{(v)})^{\top}\mathbf{P}^{(v)} = \mathbf{I},
\end{aligned}
\tag{12}
$$

where $\gamma_{max}$ is the largest eigenvalue of $\mathbf{L}^{(v)}$. In the appendix, we will provide details on how to solve Eq. (12) based on the approach outlined in (Zhang et al., 2020).

**Update Z:** Fixing $\mathbf{P}^{(v)}$, we have the following optimization problem:

$$
\begin{aligned}
\min_{\mathbf{Z}} \sum_{v=1}^{V} &\operatorname{Tr}\big(\mathbf{Z}^{\top}\mathbf{Z}\mathbf{B}^{(v)} + 2\beta\mathbf{Z}^{\top}(\mathbf{P}^{(v)})^{\top}\mathbf{L}^{(v)}\mathbf{P}^{(v)}\mathbf{Z} \\
&- 2(\mathbf{Z}^{(v)})^{\top}\mathbf{P}^{(v)}\mathbf{Z}\mathbf{B}^{(v)}\big), \\
&\text{s.t. } \mathbf{Z}^{\top}\mathbf{1} = \mathbf{1}, \mathbf{Z} \geq 0.
\end{aligned}
\tag{13}
$$

Next, we derive the optimization problem for the $j$-th column of $\mathbf{Z}$ as follows:

$$
\begin{aligned}
\min_{\mathbf{Z}_{:,j}} &\mathbf{Z}_{:,j}^{\top}\mathbf{H}\mathbf{Z}_{:,j} - 2r_j\mathbf{Z}_{:,j}, \\
&\text{s.t. } \mathbf{Z}_{:,j}^{\top}\mathbf{1} = 1, \mathbf{Z}_{:,j} \geq 0,
\end{aligned}
\tag{14}
$$

where $\mathbf{H} = \sum_{v=1}^{V}\theta_j^{(v)}\mathbf{I} + 2\beta(\mathbf{P}^{(v)})^{\top}\mathbf{L}^{(v)}\mathbf{P}^{(v)}$, $r_j = \sum_{v=1}^{V}\theta_j^{(v)}((\mathbf{Z}^{(v)})^{\top}\mathbf{P}^{(v)})_{j,:}$. Similarly, we solve Eq. (14) with the quadratic programming package.

The optimization process of EFIMVC is presented in Algorithm 1. In the appendix, we will sequentially prove the convergence of the objective function on both the client and server. Additionally, we will demonstrate that, despite independent optimization at each end, the overall objective function is still convergent.

To facilitate the identification of missing samples, we define an indicator vector $\theta^{(v)}$ as follows,

$$
\theta_j^{(v)} = \begin{cases} 1, & j\text{-th sample exists in the } v\text{-th view,} \\ 0, & \text{else.} \end{cases}
\tag{9}
$$

Note that in Eq. (8), each column of $\mathbf{Z}^{(v)}$ is independent, and only the columns corresponding to existing samples need to be optimized. For all $\theta_j^{(v)} = 1$, we optimize $\mathbf{Z}_{:,j}^{(v)}$ as follows:

$$
\begin{aligned}
\min_{\mathbf{Z}_{:,j}^{(v)}} &(\mathbf{Z}_{:,j}^{(v)})^{\top}\mathbf{W}^{(v)}\mathbf{Z}_{:,j}^{(v)} - 2t_j^v\mathbf{Z}_{:,j}^{(v)}, \\
&\text{s.t. } (\mathbf{Z}_{:,j}^{(v)})^{\top}\mathbf{1} = 1, \mathbf{Z}_{:,j}^{(v)} \geq 0,
\end{aligned}
\tag{10}
$$

where $\mathbf{W}^{(v)} = (\mathbf{A}^{(v)})^{\top}\mathbf{A}^{(v)} + \lambda\mathbf{I}$, $t_j^v = (\mathbf{X}_{:,j}^{(v)})^{\top}\mathbf{A}^{(v)} + \lambda(\mathbf{P}^{(v)}\mathbf{Z})_{:,j}^{\top}$. Eq. (10) is a standard quadratic programming problem, which we solve using existing software packages.

### 3.4.2. OPTIMIZATION IN SERVER

**Update $\mathbf{P}^{(v)}$:** Fixing $\mathbf{Z}$, $\mathbf{P}^{(v)}$ in each view is independent, we optimize it as follows:

$$
\begin{aligned}
\min_{\mathbf{P}^{(v)}} &\operatorname{Tr}\big(\beta(\mathbf{P}^{(v)})^{\top}\mathbf{L}^{(v)}\mathbf{P}^{(v)}\mathbf{Z}\mathbf{Z}^{\top} - (\mathbf{P}^{(v)})^{\top}\mathbf{Z}^{(v)}\mathbf{B}^{(v)}\mathbf{Z}^{\top}\big), \\
&\text{s.t. } (\mathbf{P}^{(v)})^{\top}\mathbf{P}^{(v)} = \mathbf{I}.
\end{aligned}
\tag{11}
$$

## 4. Experiment

### 4.1. Experimental Setup

**Employed Datasets.** We conduct experiments on seven multi-view datasets: ProteinFold, WebKB, 100Leaves, CCV, YTF10, CIFAR10, and MNIST, with detailed descriptions

*Table 2.* Clustering performance comparison of four incomplete MVC methods and four FIMVC methods on seven datasets. The best is marked in bold and underlined, the second best is marked in bold.

| Datasets | FIMVC | SCBGL | DVSAI | DAQINT | FMVC | HFMVC | FMCSC | FedMVFPC | Proposed |
|---|---|---|---|---|---|---|---|---|---|
| | | | | ACC (%) | | | | | |
| ProteinFold | 28.15 | 28.33 | **32.36** | 29.62 | 16.73 | 23.02 | 23.85 | 19.39 | **31.71** |
| WebKB | **91.51** | 75.21 | 68.64 | 78.09 | 81.78 | 57.64 | 58.27 | 75.70 | **90.64** |
| 100Leaves | 47.12 | 47.00 | 53.73 | 52.15 | 10.24 | 35.09 | 47.97 | 15.94 | **72.88** |
| CCV | 19.23 | 19.33 | **20.98** | 19.14 | - | 16.24 | **20.27** | 17.62 | 20.04 |
| YTF10 | 77.80 | 80.55 | 78.38 | 80.27 | - | 42.19 | 17.69 | 47.26 | **84.95** |
| Cifar10 | 96.56 | 96.50 | 95.83 | 95.22 | - | 57.65 | 18.73 | 69.80 | **96.62** |
| MNIST | **98.70** | 98.28 | 98.64 | 98.59 | - | 56.18 | 60.30 | 63.68 | **98.66** |
| | | | | NMI (%) | | | | | |
| ProteinFold | 36.22 | 36.15 | 39.95 | 36.75 | 19.05 | 32.05 | 32.96 | 21.59 | **40.33** |
| WebKB | 48.89 | 16.60 | 16.44 | 26.03 | 15.16 | 3.36 | 2.73 | 9.15 | **50.62** |
| 100Leaves | 70.37 | 70.36 | 74.24 | 73.93 | 30.14 | 63.77 | 70.42 | 41.33 | **84.76** |
| CCV | 14.93 | 14.89 | **15.95** | 15.14 | - | 12.74 | 15.72 | 12.40 | **15.66** |
| YTF10 | 80.22 | 81.58 | 83.79 | 82.70 | - | 39.76 | 6.09 | 47.02 | **83.86** |
| Cifar10 | 91.38 | 91.23 | 89.98 | 88.75 | - | 55.47 | 7.60 | 65.82 | **91.52** |
| MNIST | **96.17** | 95.26 | 95.95 | 95.79 | - | 53.21 | 58.90 | 60.89 | **96.04** |
| | | | | Purity (%) | | | | | |
| ProteinFold | 33.67 | 33.85 | 37.01 | 34.87 | 19.07 | 27.63 | 33.14 | 22.44 | **37.95** |
| WebKB | **91.51** | 78.12 | 79.03 | 81.92 | 81.78 | 57.04 | 61.28 | 78.12 | **90.64** |
| 100Leaves | 49.41 | 49.39 | 55.97 | 54.49 | 11.90 | 43.72 | 54.41 | 16.08 | **74.79** |
| CCV | 22.48 | 22.77 | **23.73** | 22.08 | - | 18.93 | 22.16 | 19.57 | **23.45** |
| YTF10 | 81.19 | 83.72 | 82.83 | 85.01 | - | 47.63 | 19.84 | 50.97 | **87.14** |
| Cifar10 | 96.56 | 96.46 | 95.83 | 95.22 | - | 58.44 | 19.91 | 69.89 | **96.62** |
| MNIST | **98.70** | 98.28 | 98.64 | 98.59 | - | 60.09 | 68.81 | 64.22 | **98.66** |
| | | | | Fscore (%) | | | | | |
| ProteinFold | 15.53 | 15.34 | **18.98** | 16.21 | 9.19 | 12.16 | 13.52 | 10.48 | **18.71** |
| WebKB | **88.66** | 69.02 | 64.38 | 74.34 | 79.99 | 58.62 | 59.11 | 71.44 | **86.80** |
| 100Leaves | 31.77 | 31.68 | 38.48 | 37.60 | 2.56 | 19.21 | 33.74 | 6.68 | **61.84** |
| CCV | 11.27 | 11.21 | **12.08** | 11.01 | - | 10.26 | 11.45 | 11.46 | **11.79** |
| YTF10 | 74.23 | 77.02 | 77.27 | 77.54 | - | 32.41 | 13.09 | 36.34 | **79.57** |
| Cifar10 | 93.35 | 93.18 | 91.99 | 90.85 | - | 46.00 | 13.33 | 59.03 | **93.46** |
| MNIST | **97.44** | 96.63 | 97.30 | 97.19 | - | 45.88 | 50.73 | 48.67 | **97.36** |

provided in Table 1. Following the definition in (Wang et al., 2022), we generate nine versions of each dataset with missing rates increasing in 10% increments, ensuring that no sample is entirely missing from all views.

**Compared methods.** Since no existing federated multi-view clustering methods specifically address the incomplete multi-view scenario, we compare EFIMVC with two categories of baseline methods. The first includes incomplete multi-view clustering algorithms: FIMVC (Liu et al., 2022), SCBGL (Zhao et al., 2023), DSVAI (Yu et al., 2024a), and DAQINT (Yu et al., 2024b). The second comprises federated multi-view clustering methods: FMVC (Feng et al., 2024a), HFMVC (Jiang et al., 2024), FMCSC (Chen et al., 2024), and FedMVFPC (Hu et al., 2023).

**Implementation details.** For all baselines, we follow the

parameter settings from their corresponding literature and evaluate them on datasets with different missing rates, reporting the final averaged results. Specifically, for federated multi-view clustering algorithms that cannot handle missing views, we fill the missing entries with zeros before inputting the data. Additionally, for methods that obtain final results through $k$-means, we repeat the clustering process 20 times and report the average performance to mitigate the impact of initialization randomness.

### 4.2. Performance Evaluation

We evaluate EFIMVC against eight state-of-the-art algorithms across seven datasets, as summarized in Table 2. The first four baselines are designed for incomplete multi-view clustering, while the latter four are tailored for federated learning. EFIMVC achieves competitive or superior results

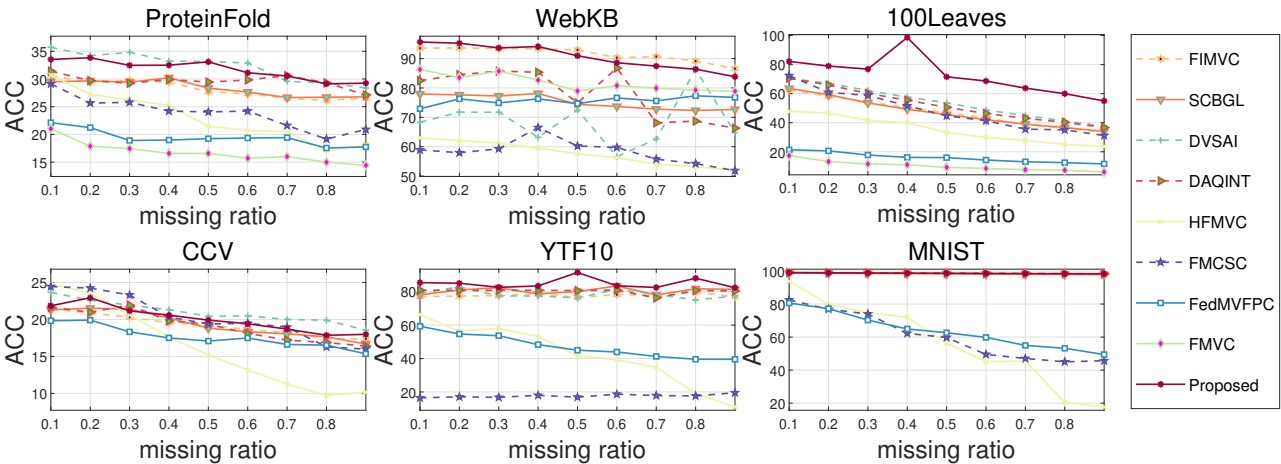

*Figure 2.* The clustering performance curves of different methods across six datasets under varying missing rates (ACC).

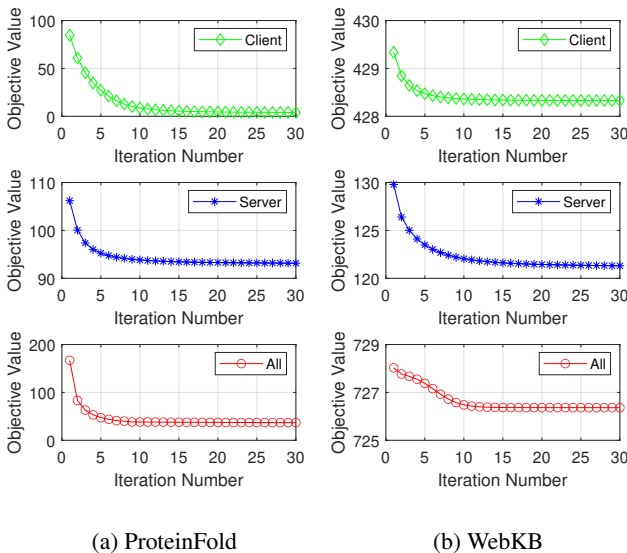

(a) ProteinFold  (b) WebKB

*Figure 3.* The objective function value curves of the client, server, and overall optimization process with respect to the number of iterations.

compared to both sets of methods, especially in handling incomplete multi-view data.

- Compared to state-of-the-art incomplete MVC methods designed for centralized storage of multi-view data, EFIMVC achieves comparable or even superior clustering performance. Specifically, on the 100Leaves and YTF10 datasets, EFIMVC outperforms the best incomplete MVC method by 35.6% and 5.5% in terms of ACC, respectively.

- Compared to four federated multi-view clustering

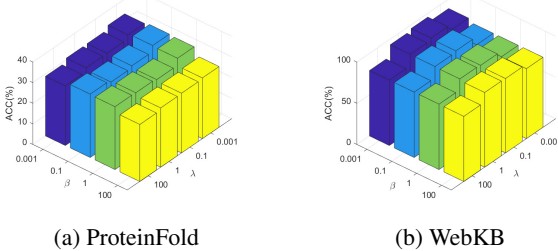

(a) ProteinFold  (b) WebKB

*Figure 4.* Sensitivity analysis of $\lambda$ and $\beta$ on ProteinFold and WebKB. Others are displayed in appendix.

methods, which struggle to handle multi-view data (requiring zero-padding for missing values during experiments), EFIMVC demonstrates superior clustering accuracy. On the ProteinFold and WebKB datasets, EFIMVC outperforms the second-best method by 32.9% and 19.7% in terms of ACC, respectively.

- In Figure 2, we compare the ACC curves of EFIMVC with other methods under varying missing view rates. EFIMVC consistently achieves superior results across all missing rates, maintaining stable accuracy even as the missing view rate increases. This highlights the robustness of EFIMVC in both incomplete data and federated learning scenarios.

### 4.3. Ablation Study

To assess the effectiveness of key components in EFIMVC, we conduct an ablation study by removing (1) the regularization term by setting $\lambda = 0$, (2) the feature alignment term by setting $\beta = 0$, and (3) anchor alignment by setting $\mathbf{P}^{(v)}$

*Table 3.* Ablation study on different modules.

| Datasets | ProteinFold | WebKB | 100Leaves | CCV | YTF10 | Cifar10 | MNIST |
|----------|-------------|-------|-----------|-----|-------|---------|-------|
| | | | ACC (%) | | | | |
| lambda=0 | 30.80 | 78.68 | 64.09 | 17.96 | 79.44 | 96.01 | 98.51 |
| beta=0 | 30.85 | 90.36 | 68.12 | 14.85 | 78.42 | 96.54 | 98.64 |
| P=I | 20.18 | 73.61 | 54.46 | 14.82 | 77.04 | 82.70 | 91.25 |
| Proposed | **31.71** | **90.64** | **72.88** | **20.04** | **84.95** | **96.62** | **98.66** |
| | | | NMI (%) | | | | |
| lambda=0 | 39.87 | 29.43 | 79.91 | 12.90 | 81.70 | 90.30 | 95.73 |
| beta=0 | 40.17 | 49.78 | 82.10 | 10.77 | 79.71 | 91.39 | 95.99 |
| P=I | 26.96 | 22.66 | 74.86 | 10.96 | 77.88 | 73.91 | 88.10 |
| Proposed | **40.33** | **50.62** | **84.76** | **15.66** | **83.86** | **91.52** | **96.04** |
| | | | Purity (%) | | | | |
| lambda=0 | 37.79 | 82.34 | 66.66 | 21.48 | 84.50 | 96.01 | 98.51 |
| beta=0 | 37.83 | 90.36 | 70.44 | 18.54 | 81.47 | 96.54 | 98.64 |
| P=I | 25.21 | 81.60 | 56.83 | 18.70 | 80.21 | 83.19 | 91.92 |
| Proposed | **37.95** | **90.64** | **74.79** | **23.45** | **87.14** | **96.62** | **98.66** |
| | | | Fscore (%) | | | | |
| lambda=0 | 17.81 | 73.21 | 50.67 | 10.34 | 74.94 | 92.28 | 97.07 |
| beta=0 | 17.79 | 86.42 | 55.60 | 9.06 | 73.51 | 93.32 | 97.32 |
| P=I | 9.01 | 69.83 | 39.62 | 9.05 | 72.42 | 73.50 | 87.56 |
| Proposed | **18.71** | **86.80** | **61.84** | **11.79** | **79.57** | **93.46** | **97.36** |

as the identity matrix. As shown in Table 3, removing any module degrades clustering performance. Specifically, removing the regularization term results in suboptimal anchor graph learning, highlighting the importance of guidance by global anchor graph. Completely removing the anchor alignment mechanism results in a significant performance drop due to misaligned anchor graph fusion on the server. In contrast, omitting the feature alignment term leads to a more gradual decline in performance.

### 4.4. Convergence Analysis

In Figure 3, we present the curves showing the objective function values on the client, on the server, and the overall objective function values as a function of the number of iterations. All three objective function values monotonically decrease and converge to stable values within 30 iterations. Notably, we observe that the overall objective function converges more quickly than the ones on the client and server, indicating reduced communication overhead.

### 4.5. Sensitivity Analysis

EFIMVC has two hyperparameters, $\lambda$ and $\beta$, with search ranges of $[0.001, 0.1, 1, 100]$. The performance of EFIMVC under different combinations of these hyperparameters are shown in Figure 4. Specifically, on ProteinFold, setting $\lambda = 0.001$ yields the best clustering performance, while on the WebKB dataset, $\lambda = 0.1$ provides the optimal results. When $\lambda$ is fixed, the algorithm's performance is not significantly affected by changes in $\beta$.

## 5. Conclusion

In conclusion, we propose EFIMVC, a novel federated multi-view clustering method that effectively addresses communication overhead, privacy preservation, and incomplete multi-view data in federated learning. By leveraging a federated optimization framework and anchor graphs for privacy, EFIMVC achieves strong clustering performance while minimizing communication costs. It also demonstrates robust performance in handling incomplete data, making it suitable for real-world applications. Experimental results validate the superiority of EFIMVC over existing methods, offering a practical solution for federated multi-view clustering.

## Acknowledgement

This work is supported by the National Science Fund for Distinguished Young Scholars of China (No. 62325604), and the National Natural Science Foundation of China (No. 62476281 and 62276271).

## Impact Statement

This paper presents work whose goal is to advance the field of Machine Learning. There are many potential societal consequences of our work, none of which we feel must be specifically highlighted here.

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

# A. Appendix

## A.1. Algorithm for Solving Eq. (12)

Define $\mathbf{C} = \gamma_{max}\mathbf{I} - \mathbf{L}^{(v)}$, $\mathbf{D} = \mathbf{Z}\mathbf{Z}^\top$ and $\mathbf{H} = \mathbf{Z}^{(v)}\mathbf{B}^{(v)}\mathbf{Z}^\top$, Eq. (12) can be rewrite as follows,

$$\max_{\mathbf{P}^{(v)}} \text{Tr}\left((\mathbf{P}^{(v)})^\top \mathbf{C}\mathbf{P}^{(v)}\mathbf{D} + (\mathbf{P}^{(v)})^\top \mathbf{H}\right),$$
$$\text{s.t. } (\mathbf{P}^{(v)})^\top \mathbf{P}^{(v)} = \mathbf{I}. \tag{15}$$

**Theorem A.1.** *Given* $\mathbf{Z} \in \mathbb{R}^{m \times n}$, $\mathbf{Z}\mathbf{Z}^\top$ *is a positive semidefinite matrix.*

*Proof.* For any vector $x \in \mathbb{R}^m$, we have

$$x^\top \left(\mathbf{Z}\mathbf{Z}^\top\right) x = x^\top \left(\sum_{j=1}^{n} \mathbf{Z}_{:,j}\mathbf{Z}_{:,j}^\top\right) x = \sum_{j=1}^{n} x^\top \mathbf{Z}_{:,j}\mathbf{Z}_{:,j}^\top x. \tag{16}$$

Since squares of real numbers are always non-negative, we have

$$x^\top \left(\mathbf{Z}\mathbf{Z}^\top\right) x = \sum_{j=1}^{n}(x^\top \mathbf{Z}_{:,j})^2 \geq 0. \tag{17}$$

Since $x^\top \left(\mathbf{Z}\mathbf{Z}^\top\right) x \geq 0$ for all $x$, the matrix $\mathbf{Z}\mathbf{Z}^\top$ is positive semi-definite. $\square$

According to (Nie et al., 2014), given two positive semidefinite matrices $\mathbf{C}$ and $\mathbf{D}$, the function $f(\mathbf{P}^{(v)}) = \text{Tr}\left((\mathbf{P}^{(v)})^\top \mathbf{C}\mathbf{P}^{(v)}\mathbf{D} + (\mathbf{P}^{(v)})^\top \mathbf{H}\right)$ is convex. The optimal solution of $\max_{(\mathbf{P}^{(v)})^\top \mathbf{P}^{(v)}=\mathbf{I}} f(\mathbf{P}^{(v)})$ is equal to solve the following problem,

$$\max_{\mathbf{P}^{(v)},\mathbf{Q}} \text{Tr}\left((\mathbf{P}^{(v)})^\top \mathbf{Q}\right),$$
$$\text{s.t. } (\mathbf{P}^{(v)})^\top \mathbf{P}^{(v)} = \mathbf{I}, \tag{18}$$

where $\mathbf{Q} = f'(\mathbf{P}^{(v)}) = 2\mathbf{C}\mathbf{P}^{(v)}\mathbf{D} + \mathbf{H}$. Eq. (18) is easy to solved by iteratively optimizing $\mathbf{P}^{(v)}$ and $\mathbf{Q}$ until converge.

## A.2. Theoretical Analysis for Convergence

In this section, we first analyse the convergence of the optimization process on the client side, followed by the convergence analysis on the server side, and finally, we provide the overall convergence analysis of the algorithm.

First, the objective function of the $v$-th client is defined as follows:

$$\mathcal{J}_c\left(\mathbf{A}^{(v)}, \mathbf{Z}^{(v)}\right) = \left\|\mathbf{X}^{(v)}\mathbf{G}^{(v)} - \mathbf{A}^{(v)}\mathbf{Z}^{(v)}\mathbf{G}^{(v)}\right\|_{\mathbf{F}}^2 + \lambda\left\|\mathbf{P}^{(v)}\mathbf{Z}\mathbf{G}^{(v)} - \mathbf{Z}^{(v)}\mathbf{G}^{(v)}\right\|_{\mathbf{F}}^2, \tag{19}$$

When solving Eq. (19), we iteratively optimize each variable while keep others fixed. The optimal variables in the $t_c$ iteration are defined as $\mathbf{A}_{t_c}^{(v)}, \mathbf{Z}_{t_c}^{(v)}$. In the $t_c + 1$ iteration, we optimize $\mathbf{A}^{(v)}$ with fixed $\mathbf{Z}_{t_c}^{(v)}$. We have

$$\mathcal{J}_c\left(\mathbf{A}_{t_c}^{(v)}, \mathbf{Z}_{t_c}^{(v)}\right) \geq \mathcal{J}_c\left(\mathbf{A}_{t_c+1}^{(v)}, \mathbf{Z}_{t_c}^{(v)}\right). \tag{20}$$

Then we optimize $\mathbf{Z}^{(v)}$ with fixed $\mathbf{A}_{t_c+1}^{(v)}$. We have

$$\mathcal{J}_c\left(\mathbf{A}_{t_c+1}^{(v)}, \mathbf{Z}_{t_c}^{(v)}\right) \geq \mathcal{J}_c\left(\mathbf{A}_{t_c+1}^{(v)}, \mathbf{Z}_{t_c+1}^{(v)}\right). \tag{21}$$

Along with Eq. (20), we have

$$\mathcal{J}_c\left(\mathbf{A}_{t_c}^{(v)}, \mathbf{Z}_{t_c}^{(v)}\right) \geq \mathcal{J}_c\left(\mathbf{A}_{t_c+1}^{(v)}, \mathbf{Z}_{t_c+1}^{(v)}\right), \tag{22}$$

which indicates that the objective function of the $v$-th client is monotonically decreasing as the number of iterations increases. Moreover, since the objective function is lower-bounded by 0, the optimization process on the $v$-th client is guaranteed to converge.

Defining the objective function of the server as follows:

$$\mathcal{J}_s\left(\mathbf{Z}, \mathbf{P}^{(v)}\right) = \sum_{v=1}^{V} \left\|\mathbf{P}^{(v)}\mathbf{Z}\mathbf{G}^{(v)} - \mathbf{Z}^{(v)}\mathbf{G}^{(v)}\right\|_{\mathbf{F}}^2 + \beta \left\|(\mathbf{P}^{(v)}\mathbf{Z})_{i,:} - (\mathbf{P}^{(v)}\mathbf{Z})_{j,:}\right\|_2^2 \mathbf{S}_{i,j}^{(v)}. \tag{23}$$

Similar to the previous analysis, let $\mathbf{Z}_{t_s}$ and $\mathbf{P}_{t_s}^{(v)}$ be the optimal variables at the $t_s$-th iteration. Then, we can derive that

$$\mathcal{J}_s\left(\mathbf{Z}_{t_s}, \mathbf{P}_{t_s}^{(v)}\right) \geq \mathcal{J}_s\left(\mathbf{Z}_{t_s+1}, \mathbf{P}_{t_s+1}^{(v)}\right). \tag{24}$$

The objective function value also decreases monotonically with the number of iterations. Moreover, since Eq. (23) is lower-bounded by 0, the optimization process on the server is guaranteed to converge.

Finally, we defined the overall function of EFIMVC by combining Eq. (19) and Eq. (23) as follows,

$$\mathcal{J}_o\left(\mathbf{A}^{(v)}, \mathbf{Z}^{(v)}, \mathbf{Z}, \mathbf{P}^{(v)}\right) = \sum_{v=1}^{V} \left\|\mathbf{P}^{(v)}\mathbf{Z}\mathbf{G}^{(v)} - \mathbf{Z}^{(v)}\mathbf{G}^{(v)}\right\|_{\mathbf{F}}^2 + \sum_{v=1}^{V} \frac{1}{\lambda} \left\|\mathbf{X}^{(v)}\mathbf{G}^{(v)} - \mathbf{A}^{(v)}\mathbf{Z}^{(v)}\mathbf{G}^{(v)}\right\|_{\mathbf{F}}^2 \\ + \beta \left\|(\mathbf{P}^{(v)}\mathbf{Z})_{i,:} - (\mathbf{P}^{(v)}\mathbf{Z})_{j,:}\right\|_2^2 \mathbf{S}_{i,j}^{(v)}. \tag{25}$$

After the $t_o$-th iteration optimization, the optimal variables are defined as $\mathbf{A}_{t_o}^{(v)}, \mathbf{Z}_{t_o}^{(v)}, \mathbf{Z}_{t_o}$ and $\mathbf{P}_{t_o}^{(v)}$. According to Algorithm 1, in the $t_o + 1$-th round, $\mathbf{Z}_{t_o}$ and $\mathbf{P}_{t_o}^{(v)}$ are transmitted from the server to the clients for the optimization of $\mathbf{A}^{(v)}$ and $\mathbf{Z}^{(v)}$. After the alternating optimization process on each client, $\mathbf{A}_{t_o+1}^{(v)}$ and $\mathbf{Z}_{t_o+1}^{(v)}$ are obtained. Based on Eq. (22), we have

$$\mathcal{J}_o\left(\mathbf{A}_{t_o}^{(v)}, \mathbf{Z}_{t_o}^{(v)}, \mathbf{Z}_{t_o}, \mathbf{P}_{t_o}^{(v)}\right) - \mathcal{J}_o\left(\mathbf{A}_{t_o+1}^{(v)}, \mathbf{Z}_{t_o+1}^{(v)}, \mathbf{Z}_{t_o}, \mathbf{P}_{t_o}^{(v)}\right) \\ = \frac{1}{\lambda}\mathcal{J}_c\left(\mathbf{A}_{t_o}^{(v)}, \mathbf{Z}_{t_o}^{(v)}\right) - \frac{1}{\lambda}\mathcal{J}_c\left(\mathbf{A}_{t_o+1}^{(v)}, \mathbf{Z}_{t_o+1}^{(v)}\right) \geq 0. \tag{26}$$

Therefore,

$$\mathcal{J}_o\left(\mathbf{A}_{t_o}^{(v)}, \mathbf{Z}_{t_o}^{(v)}, \mathbf{Z}_{t_o}, \mathbf{P}_{t_o}^{(v)}\right) \geq \mathcal{J}_o\left(\mathbf{A}_{t_o+1}^{(v)}, \mathbf{Z}_{t_o+1}^{(v)}, \mathbf{Z}_{t_o}, \mathbf{P}_{t_o}^{(v)}\right). \tag{27}$$

Then the optimal $\mathbf{A}_{t_o+1}^{(v)}$ and $\mathbf{Z}_{t_o+1}^{(v)}$ are input to the server for optimizing $\mathbf{Z}$ and $\mathbf{P}^{(v)}$. Based on Eq. (24), we have

$$\mathcal{J}_o\left(\mathbf{A}_{t_o+1}^{(v)}, \mathbf{Z}_{t_o+1}^{(v)}, \mathbf{Z}_{t_o}, \mathbf{P}_{t_o}^{(v)}\right) - \mathcal{J}_o\left(\mathbf{A}_{t_o+1}^{(v)}, \mathbf{Z}_{t_o+1}^{(v)}, \mathbf{Z}_{t_o+1}, \mathbf{P}_{t_o+1}^{(v)}\right) \\ = \mathcal{J}_s\left(\mathbf{Z}_{t_o}, \mathbf{P}_{t_o}^{(v)}\right) - \mathcal{J}_s\left(\mathbf{Z}_{t_o+1}, \mathbf{P}_{t_o+1}^{(v)}\right) \geq 0, \tag{28}$$

which means

$$\mathcal{J}_o\left(\mathbf{A}_{t_o+1}^{(v)}, \mathbf{Z}_{t_o+1}^{(v)}, \mathbf{Z}_{t_o}, \mathbf{P}_{t_o}^{(v)}\right) \geq \mathcal{J}_o\left(\mathbf{A}_{t_o+1}^{(v)}, \mathbf{Z}_{t_o+1}^{(v)}, \mathbf{Z}_{t_o+1}, \mathbf{P}_{t_o+1}^{(v)}\right). \tag{29}$$

By combining Eq. (27) and Eq. (29), we can derive that,

$$\mathcal{J}_o\left(\mathbf{A}_{t_o}^{(v)}, \mathbf{Z}_{t_o}^{(v)}, \mathbf{Z}_{t_o}, \mathbf{P}_{t_o}^{(v)}\right) \geq \mathcal{J}_o\left(\mathbf{A}_{t_o+1}^{(v)}, \mathbf{Z}_{t_o+1}^{(v)}, \mathbf{Z}_{t_o+1}, \mathbf{P}_{t_o+1}^{(v)}\right). \tag{30}$$

## A.3. Supplementary Experiments

### A.3.1. MORE IMPLEMENTATION DETAILS

The parameter settings in EFIMVC are grid search. For both $\lambda$ and $\beta$, we search them in $[0.001, 0.1, 1, 100]$. To evaluate all methods, we employ four widely used metrics, including Accuracy (ACC), Normalized Mutual Information (NMI), Purity, and Fscore. All experiments are conducted on a computer with Intel Core i9-10900X CPU and 64G RAM.

### A.3.2. DETAILS OF EMPLOYED DATASETS

- ProteinFold [1]: A biological dataset containing structural representations of protein folding processes.

- WebKB [2]: A text dataset collected from webpages of four universities, where views correspond to content and citations.

- 100Leaves [3]: A leaf species classification dataset containing multi-view representations corresponding to shape descriptor, fine scale margin and texture histogram.

- CCV [4]: A video dataset consisting of different types of features extracted from frames, including SIFT, GIST, and deep features.

- YTF10 [5]: A subset of the YouTubeFaces dataset, containing facial videos with multiple feature extraction methods applied.

- CIFAR10 [6]: A well-known image dataset with different views extracted by ResNet18, ResNet50, and DenseNet121.

- MNIST [7]: A handwritten digit dataset where views generated by different feature extractor.

### A.3.3. DETAILS OF COMPARED METHODS

To evaluate the performance of EFIMVC, we compared it against eight state-of-the-art methods, categorized into two groups: incomplete multi-view clustering methods and federated multi-view clustering methods.

**Incomplete multi-view clustering methods:**

- FIMVC (Liu et al., 2022): This method introduces view-independent anchors to enhance clustering efficiency while preserving multi-view consistency. By constructing an anchor graph rather than a full pairwise graph, it reduces computational complexity and scales well to large datasets.

- SCBGL (Zhao et al., 2023): It uses bipartite graph learning to construct consensus anchor matrices, enabling effective feature alignment across views. A self-completion mechanism ensures robustness to missing views while maintaining low computational overhead.

- DSVAI (Yu et al., 2024a): It proposes diverse view-shared anchors with multiple dimensions and sizes to better represent multi-view data. A linear integration strategy ensures computational efficiency while improving clustering accuracy in large-scale scenarios.

- DAQINT (Yu et al., 2024b): This method introduces an adaptive weighting mechanism to assign varying numbers of anchors per view, eliminating manual tuning. It improves the flexibility of anchor-based clustering while balancing computational complexity and clustering performance.

**Federated multi-view clustering methods:**

- FMVC (Feng et al., 2024a): It integrates matrix factorization and K-Means clustering within a federated learning framework, enabling collaborative clustering without sharing raw data. The method preserves data privacy while maintaining strong clustering performance.

- HFMVC (Jiang et al., 2024): This method addresses heterogeneity in federated multi-view clustering by employing contrastive learning and local-global model alignment. It effectively adapts to both IID and non-IID data distributions, ensuring more stable clustering results.

---

[1] http://mkl.ucsd.edu/dataset/protein-fold-prediction/
[2] http://www.cs.umd.edu/sen/lbc-proj/LBC.html
[3] https://www.archive.ics.uci.edu/dataset/241
[4] https://www.ee.columbia.edu/ln/dvmm/CCV/
[5] https://www.micc.unifi.it/resources/datasets/e-ytf/
[6] http://www.cs.toronto.edu/kriz/cifar.html
[7] http://yann.lecun.com/exdb/mnist/

- FMCSC (Chen et al., 2024): It focuses on hybrid-view heterogeneity by leveraging contrastive learning to align single-view and multi-view clients. A global-specific weighting aggregation strategy enhances consistency across distributed clients, improving clustering quality.

- FedMVFPC (Hu et al., 2023): This method extends fuzzy clustering into a federated setting by introducing consensus prototype learning. It ensures stability in clustering performance across federated clients while maintaining privacy-preserving optimization.

### A.3.4. CLUSTERING RESULTS ON MORE METRICS

Beyond the ACC results reported in the main paper, we present additional evaluation metrics including NMI, purity, and Fscore. As shown in Figures 5, 6, and 7, EFIMVC consistently achieves superior results across all datasets and missing rates. The improvements are particularly significant in high-missing-rate scenarios, demonstrating the robustness of our approach in handling incomplete multi-view data.

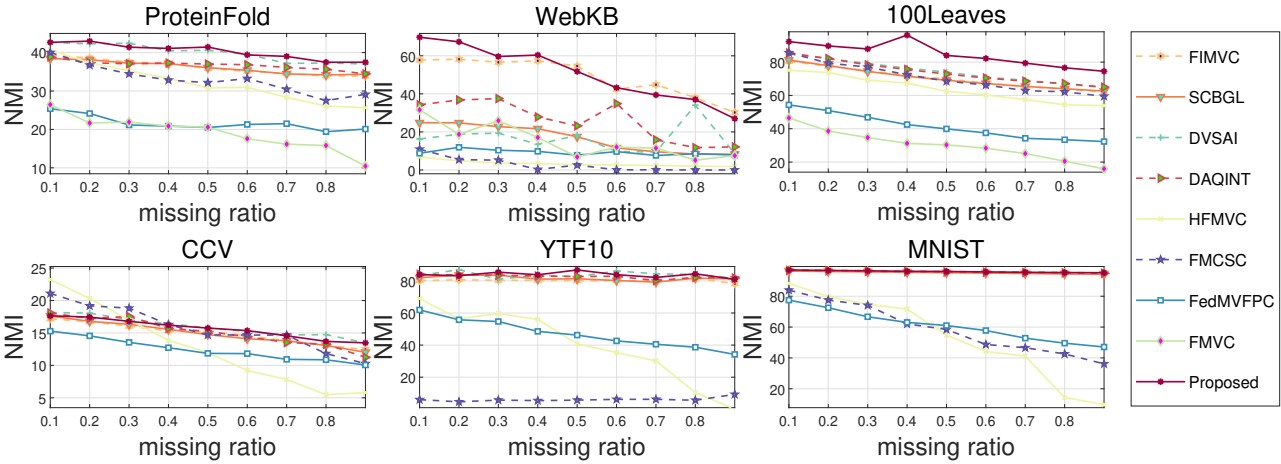

*Figure 5.* The clustering performance curves of different methods across six datasets under varying missing rates (NMI).

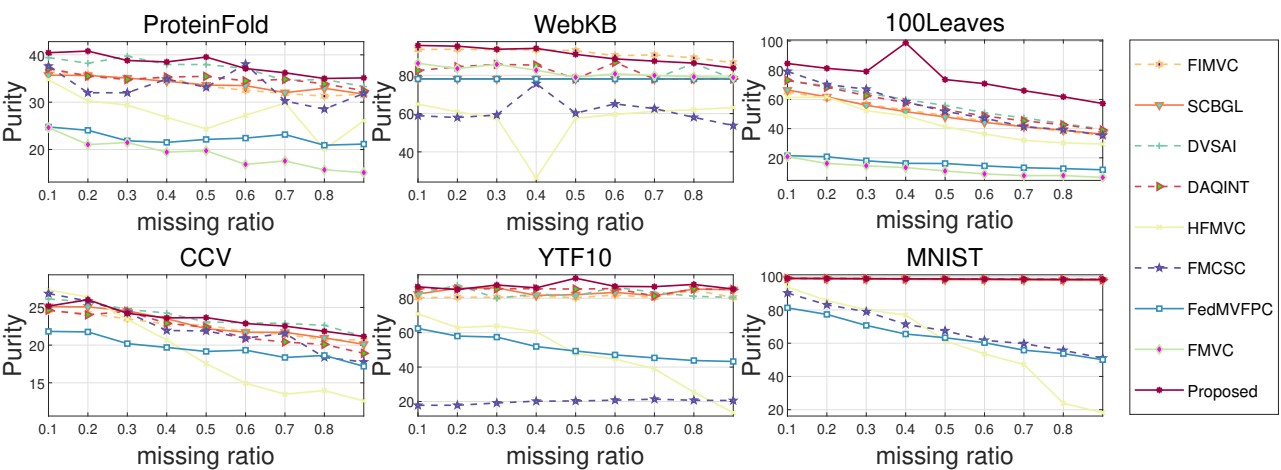

*Figure 6.* The clustering performance curves of different methods across six datasets under varying missing rates (Purity).

### A.3.5. PARAMETER ANALYSIS ON MORE DATASETS

To further investigate the impact of hyperparameters, we analyse the performance of EFIMVC under varying values of $\lambda$ and $\beta$ on more datasets. Figure 8 illustrates how different parameter settings affect clustering performance across additional

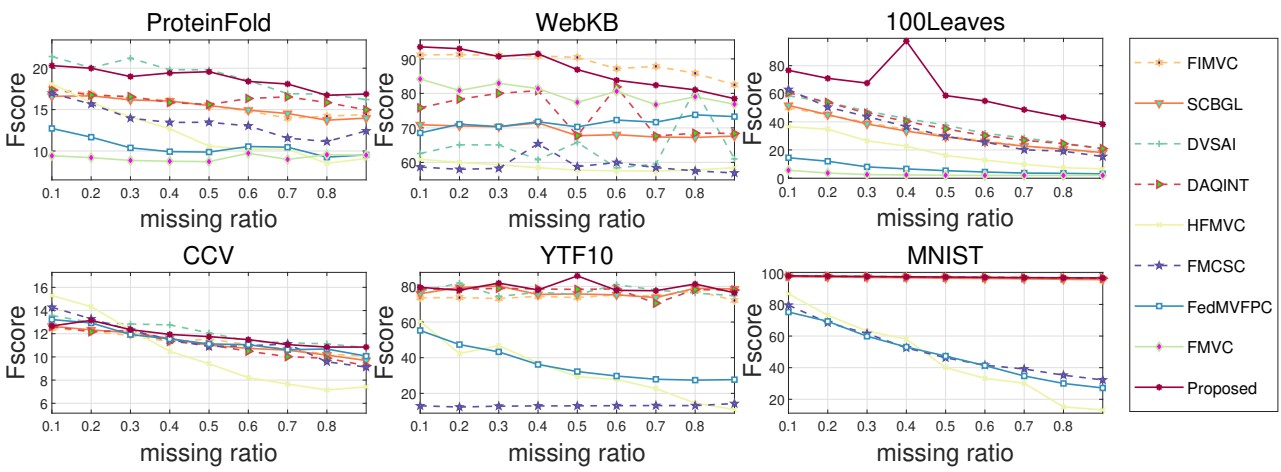

*Figure 7.* The clustering performance curves of different methods across six datasets under varying missing rates (Fscore).

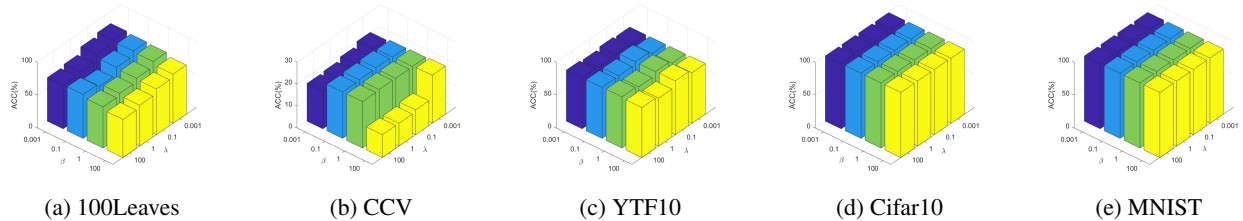

| (a) 100Leaves | (b) CCV | (c) YTF10 | (d) Cifar10 | (e) MNIST |

*Figure 8.* Sensitivity analysis of $\lambda$ and $\beta$ on other datasets.

datasets, including 100Leaves, CCV, YTF10, CIFAR10, and MNIST. The results indicate that while EFIMVC is relatively stable to changes in $\beta$, setting $\lambda$ too high or too low can negatively impact clustering accuracy. Dataset-specific tuning is thus recommended for optimal results.

