# OpenReview forum: "Efficient Federated Incomplete Multi-View Clustering"
_ICML.cc/2025/Conference — ICML 2025 poster_

### Official Review · Reviewer_NJFx · 2025-03-12

**Overall Recommendation:** 4

**Summary:**

The work presents EFIMVC, an anchor-based federated MVC method employing view-specific/shared anchor graphs and alignment matrices. While the technical approach shows some novelty, fundamental design choices lack justification, and experimental comparisons appear skewed.

**Claims And Evidence:**

The clustering performance does not demonstrate superiority.

**Essential References Not Discussed:**

None.

**Experimental Designs Or Analyses:**

I have doubts about the fairness of setting different comparison methods.

**Methods And Evaluation Criteria:**

The application of anchor graphs in federated settings is novel. The computational efficiency through anchor sampling is practical.

**Other Comments Or Suggestions:**

None.

**Other Strengths And Weaknesses:**

Strengths
The application of anchor graphs in federated settings is novel. The computational efficiency through anchor sampling is practical.

Weaknesses
1. Most anchor-based MVC methods enforce orthogonality constraints on anchor matrices to ensure diversity. The authors remove this constraint without justification. What performance changes would occur if orthogonality constraints were reinstated?
2. Previous methods typically apply k-means directly on learned embeddings. The choice to cluster left singular vectors of Z in Algorithm 1 lacks theoretical justification. Why not use standard spectral clustering?
3. The huge performance gaps between the first four methods and the remaining methods in Table 2 suggest unfair experimental settings. Were all methods given equal hyperparameter tuning efforts?

**Questions For Authors:**

Existing MVC methods optimize client/server variables under a unified objective to ensure convergence. How does EFIMVC guarantee convergence with separate client/server objectives?

**Relation To Broader Scientific Literature:**

Based on incomplete multi-view clustering and federal learning, this paper presents a new approach to handle federal incomplete multi-view data.

**Theoretical Claims:**

Yes.

---

> ### Author Rebuttal · Authors · 2025-04-01
>
> **We sincerely thank Reviewer NJFx for the thorough and constructive review. We provide point-by-point responses to the questions raised as follows:**
>
> ---
>
> **Q1:** Most anchor-based MVC methods enforce orthogonality constraints on anchor matrices to ensure diversity. The authors remove this constraint without justification. What performance changes would occur if orthogonality constraints were reinstated?
>
> **A1:** We sincerely appreciate your question. While many anchor-based methods enforce orthogonality to promote anchor diversity, we intentionally relax this constraint for two key reasons:
>
> We aim to learn anchors that are representative of each category. However, in practice, the number of anchors typically exceeds the number of categories. Under such circumstances, it is unnecessary to enforce distinctness among all anchors - anchors from the same category may exhibit similarity. Imposing strict orthogonality constraints would be overly restrictive for this purpose.
>
> Besides, our ablation studies as follows show that removing orthogonality constraints actually improves clustering accuracy across all datasets, likely because it allows for more flexible anchor representations that better capture relationships in partial observation scenarios.
>
> | **Datasets** | **ProteinFold** | **WebKB** | **100Leaves** |  **CCV**  | **Cifar10** |
> |:------------:|:---------------:|:---------:|:-------------:|:---------:|:-----------:|
> |              |                 |    ACC    |               |           |             |
> |     Ours     |    **31.71**    | **90.64** |   **72.88**   | **20.04** |  **96.62**  |
> |  Orthogonal  |      29.39      |   71.74   |     61.85     |   15.47   |    95.09    |
> |||
>
> ---
>
> **Q2:** Previous methods typically apply k-means directly on learned embeddings. The choice to cluster left singular vectors of $\mathbf{Z}$ in Algorithm 1 lacks theoretical justification. Why not use standard spectral clustering?
>
> **A2:** We sincerely thank you for the problem. Standard spectral clustering can only operate on a complete $n\times n$ similarity matrix and cannot directly process the anchor graph $\mathbf{Z}$. According to [1], $\mathbf{Z}^\top \mathbf{Z}$ can be interpreted as reconstructing the full similarity matrix from the anchor graph $\mathbf{Z}$. Notably, performing $k$-means on the right singular vectors of $\mathbf{Z}$ is theoretically equivalent to applying standard spectral clustering to $\mathbf{Z}^\top \mathbf{Z}$.
>
> [1] Kang et al. Large-scale multi-view subspace clustering in linear time. AAAI 2020.
>
> ---
>
> **Q3:** The huge performance gaps between the first four methods and the remaining methods in Table 2 suggest unfair experimental settings. Were all methods given equal hyperparameter tuning efforts?
>
> **A3:** We sincerely thank you for the question. The first four methods evaluated in Table 2 are centralized incomplete multi-view clustering algorithms. Compared to the subsequent four federated MVC methods, these centralized approaches hold inherent advantages in terms of full data accessibility and built-in mechanisms for handling missing values, thereby achieving superior clustering performance on incomplete multi-view datasets.
>
> In contrast, our proposed method not only processes distributed multi-view data but also incorporates a dedicated incomplete data handling mechanism. As a result, it achieves comparable performance to state-of-the-art centralized incomplete MVC algorithms while significantly outperforming existing federated MVC methods.
>
> Besides, for all baseline methods, we strictly followed the parameter configurations reported in their original papers. Notably, for federated MVC methods, all missing values were imputed with zeros to ensure a fair comparison. We will add these implementation details to the final version.
>
> ---
>
> **Q4:** Existing MVC methods optimize client/server variables under a unified objective to ensure convergence. How does EFIMVC guarantee convergence with separate client/server objectives?
>
> **A4:** We sincerely appreciate the comment. In contrast to conventional approaches that optimize server and client variables under a unified objective function, we propose a decoupled optimization framework that separates these two components. This innovation effectively reduces the communication overhead caused by frequent interactions. We further provide both theoretical and experimental convergence analyses for the decoupled objectives relative to the global objective function.
>
> Specifically, Figure 3 illustrates the monotonic decrease of objective values (for server-side, client-side, and global objectives) with increasing iterations, demonstrating their eventual convergence to stable values. Additionally, in Appendix A.2, we present a rigorous theoretical analysis of convergence properties for each component.

---

> > ### Comment · Reviewer_NJFx · 2025-04-03
> >
> > Thanks for the authors' efforts. After reading the rebuttal, my concerns have been addressed, and I will increase my score.

---

### Official Review · Reviewer_3S7F · 2025-03-13

**Overall Recommendation:** 3

**Summary:**

This paper proposes a novel framework called EFIMVC, addressing key challenges in federated multi-view clustering such as high communication overhead, limited privacy protection, and poor handling of missing views. EFIMVC introduces a localized optimization strategy that significantly reduces communication costs while ensuring theoretical convergence. It employs both view-specific and shared anchor graphs as communication variables to enhance privacy by avoiding the transmission of sensitive embeddings. The framework also features a dual-anchor alignment mechanism to mitigate anchor graph misalignment during graph fusion, improving clustering robustness.

**Claims And Evidence:**

Yes

**Essential References Not Discussed:**

None

**Experimental Designs Or Analyses:**

The experimental design and analysis in the paper are generally sound and well-justified. The authors have taken appropriate steps to validate the performance of EFIMVC across diverse datasets and conditions. However, further exploration of computational efficiency could provide additional insights into the efficiency of the method.

**Methods And Evaluation Criteria:**

The method is relatively applicable.

**Other Comments Or Suggestions:**

The excessive mathematical notation severely impacts readability. A notation table should be added.

**Other Strengths And Weaknesses:**

Strengths:
Novelty: The paper proposes a novel federated multi-view clustering framework, the introduction of a dual-anchor alignment mechanism to mitigate anchor graph misalignment during graph fusion is a unique contribution.
Significance: The authors' proposed local optimization mechanism enhances communication efficiency, enabling the model to handle large-scale datasets. Moreover, the use of anchor graphs for communication improves user privacy.
Clarity: The paper presents the methodology in a clear and structured manner, with detailed descriptions of the optimization process, algorithm design, and theoretical analysis.  This clarity makes it easier for readers to understand.


Weaknesses:
1.	The authors claim their method solves three key challenges (communication overhead, privacy preservation, incomplete views) but fail to explicitly demonstrate how each component of the framework addresses these issues in the method part.
2.	The clustering performance does not demonstrate superiority. As shown in Table 2, EFIMVC underperforms state-of-the-art methods on ProteinFold, WebKB, CCV and MNIST datasets.
3.	In experiments, the federated baselines perform poorly compared to centralized methods. Were these baselines adapted fairly to handle missing view?

**Questions For Authors:**

What motivated the specific construction strategy for anchor similarity matrix in Eq. (3)? Would alternative similarity measures like cosine distance yield different results?

**Relation To Broader Scientific Literature:**

The authors point out that existing FMVC methods still face challenges such as high communication overhead, limited privacy protection, and ineffective handling of missing views. These issues indeed exist in federated multi-view scenarios. To address the problem of limited privacy protection, the authors use anchor graphs as communication variables, employing both view-specific and shared anchor graphs to avoid the direct transmission of sensitive embeddings, thereby enhancing privacy protection. However, the authors do not provide detailed explanations on how the model handles missing data. Although large datasets were used and improvements in clustering performance were observed, the experimental section lacks analysis of time and space complexity and does not include experiments on time efficiency to validate its effectiveness.

**Theoretical Claims:**

There is relevant theoretical verification.

---

> ### Author Rebuttal · Authors · 2025-04-01
>
> **We sincerely thank Reviewer 3S7F for the thorough and constructive review. We provide point-by-point responses to the questions raised as follows:**
>
> ---
>
> **Q1:** The authors claim their method solves three key challenges (communication overhead, privacy preservation, incomplete views) but fail to explicitly demonstrate how each component of the framework addresses these issues in the method part.
>
> **A1:** We sincerely appreciate your question. Our framework explicitly addresses the three challenges through dedicated components:
> 1. Communication Overhead: By decoupling optimization variables into server-side and client-side, we significantly reduce the variable transmission costs inherent in conventional frameworks, where optimization is performed alternately between the server and clients.
> 2. Privacy Preservation: The proposed framework employs anchor graphs as communication variables, avoiding the transmission of sensitive embeddings or sample-level similarity matrices, thereby reducing privacy risks.
> 3. Missing-View Handling: We learn partial anchor graphs with available data on each client via Eq. (2), and integrates them into a consistent full anchor graph on server-side via Eq. (5).
>
> ---
>
> **Q2:** The clustering performance does not demonstrate superiority. As shown in Table 2, EFIMVC underperforms state-of-the-art methods on ProteinFold, WebKB, CCV and MNIST datasets.
>
> **A2:** We sincerely thank you for the critical perspective. While centralized SOTA methods (FIMVC, SCBGL, DVSAI, DAQINT) achieve marginally higher accuracy on some datasets, they fundamentally violate federated constraints. In contrast, our federated framework achieves comparable or superior performance to these centralized SOTA methods while operating under strict distributed data storage protocols. This demonstrates our method’s unique capability to balance clustering accuracy with federated learning requirements. We will emphasize this distinction in the final version.
>
> ---
>
> **Q3:** In experiments, the federated baselines perform poorly compared to centralized methods. Were these baselines adapted fairly to handle missing view?
>
> **A3:** We sincerely appreciate the comment. First, federated multi-view clustering methods inherently underperform centralized approaches because they cannot simultaneously access all view data. Second, since existing federated MVC methods lack the capability to handle missing multi-view data, we filled all missing values with zeros in our experiments. This also leads to slightly inferior performance compared to specialized missing MVC algorithms with built-in imputation mechanisms. Notably, as the first federated incomplete MVC method proposed, our approach ensures a fair experimental comparison by benchmarking against both state-of-the-art centralized missing MVC methods and existing federated MVC methods.
>
> ---
>
> **Q4:** The excessive mathematical notation severely impacts readability. A notation table should be added.
>
> **A4:** We sincerely appreciate the constructive suggestion. We will add a notation table in the final version. Specially, the main used notations through our paper is summarized as follows:
>
> | Notation       | Meaning |
> |--------------|---------|
> | $\mathbf{A}^{(v)}$  | Anchor matrix|
> | $\mathbf{Z}^{(v)}$ | Anchor graph |
> | $\mathbf{Z}$  | Consistent anchor graph    |
> | $\mathbf{G}^{(v)}$   | Incomplete indicator matrix  |
> | $\mathbf{P}^{(v)}$| Alignment matrix  |
> | $\mathbf{S}^{(v)}$ | Anchor similarity matrix    |
> | $\mathbf{L}^{(v)}$  | Laplacian matrix  |
> | ||
>
>
> ---
>
> **Q5:** What motivated the specific construction strategy for anchor similarity matrix in Eq. (3)? Would alternative similarity measures like cosine distance yield different results?
>
> **A5:** We sincerely thank you for the insightful question. The anchor similarity matrix quantifies pairwise distances between all anchors, serving as the foundation for subsequent server-side anchor structure alignment. In Eq. (3), we compute the distance between each anchor pair using the ℓ2-norm (Euclidean distance), though this metric can be substituted with alternative methods (e.g., cosine similarity). To evaluate the impact of distance metrics, we replace the L2-norm with cosine similarity and compare the results with existing approaches, as shown in the table below.
>
> | **Datasets** | **ProteinFold** | **WebKB** | **100Leaves** |  **CCV**  | **Cifar10** |
> |:------------:|:---------------:|:---------:|:-------------:|:---------:|:-----------:|
> |              |                 |    ACC    |               |           |             |
> |    ℓ2-norm   |    **31.71**    | **90.64** |   **72.88**   | **20.04** |  **96.62**  |
> |    Cosine    |      31.45      |   90.44   |     70.23     |   15.17   |    96.39    |
> | ||
>
> From the experimental results, ℓ2-norm performs slightly better than cosine similarity. We will include this ablation experiment in the final version.

---

### Official Review · Reviewer_9Soy · 2025-03-14

**Overall Recommendation:** 5

**Summary:**

Tha authors propose a novel federated incomplete multi-view clustering method named EFIMVC. By introducing localized optimization with anchor graphs and dual-alignment mechanisms, the propposed method reduce communication costs while preserving privacy. Experiments on seven datasets demonstrate the superiorty of EFIMVC.

**Claims And Evidence:**

Yes. The claims made in the submission are supported by clear and convincing evidence.

**Essential References Not Discussed:**

Related work are referenced in this paper.

**Experimental Designs Or Analyses:**

Yes.

**Methods And Evaluation Criteria:**

Yes. The proposed methods make sense for the federal multi-view scenario.

**Other Comments Or Suggestions:**

See weakness.

**Other Strengths And Weaknesses:**

The practical focus on incomplete view scenarios is valuable. The anchor graph communication strategy effectively balances efficiency and information preservation. The paper is well-written. While there are some issues need to be addressed:
I. The paper uses excessive mathematical symbols without proper definitions. For instance, L in Eq.13 is never explained.
II. The "dual-anchor alignment" concept remains vague. How does it differ from single-level alignment in [1]?
III. The missing view definition is ambiguous. Is it sample-level missing (some samples lack views) or feature-level missing (entire view channels unavailable)?
IIII. Table 2 shows empty results for FMVC on four datasets. Is this due to implementation errors or intentional omission?
[1] Align then Fusion: Generalized Large-scale Multi-view Clustering with Anchor Matching Correspondences. NeurIPS 2022.

**Questions For Authors:**

See weakness.

**Relation To Broader Scientific Literature:**

EFIMVC is related to the federal multi-view methods.

**Theoretical Claims:**

Yes.The proofs are correct.

---

> ### Author Rebuttal · Authors · 2025-04-01
>
> **We sincerely thank Reviewer 9Soy for the thorough and constructive review. We provide point-by-point responses to the questions raised as follows:**
>
> ---
>
> **Q1:** The paper uses excessive mathematical symbols without proper definitions. For instance, $\mathbf{L}$ in Eq. (13) is never explained.
>
> **A1:** We sincerely appreciate your feedback. The symbol $\mathbf{L}^{(v)}$ in Eq. (13) denotes the Laplacian matrix derived from the anchor similarity matrix $\mathbf{S}^{(v)}$ of the $v$-th view, where $\mathbf{L}^{(v)} = \mathbf{D}^{(v)} - \mathbf{S}^{(v)}$ and $\mathbf{D}^{(v)}$ is the degree matrix. We acknowledge the oversight in explicitly defining this notation and will comprehensively explain all mathematical symbols in Appendix to ensure full clarity in the final version.
>
> ---
>
> **Q2:** The "dual-anchor alignment" concept remains vague. How does it differ from single-level alignment in [1]?
>
> [1] Align then Fusion: Generalized Large-scale Multi-view Clustering with Anchor Matching Correspondences. NeurIPS 2022.
>
> **A2:** We sincerely thank you for this insightful comparison. While [1] achieves alignment through sequential feature alignment and structural alignment, our dual-anchor alignment **unifies these two objectives** into a single optimization framework in Eq. (5). Specifically:
> - Feature alignment (anchor embedding matching):
> $$\left\|\left\|\mathbf{P}^{(v)}\mathbf{Z}\mathbf{G}^{(v)} - \mathbf{Z}^{(v)}\mathbf{G}^{(v)} \right\|\right\|_{\mathbf{F}}^2$$
> - Structural alignment (anchor graph topology preservation):
> $$\operatorname{Tr}\left((\mathbf{P}^{(v)})^\top\mathbf{L}^{(v)}\mathbf{P}^{(v)}(\mathbf{Z}\mathbf{Z}^\top)\right)$$
>
> Besides, our method introduces **missing-view robustness** through the indicator matrix $\mathbf{G}^{(v)}$, enabling alignment even when partial data observations are absent across views. This differs fundamentally from [1], which assumes complete data availability.
>
> ---
>
> **Q3:** The missing view definition is ambiguous. Is it sample-level missing (some samples lack views) or feature-level missing (entire view channels unavailable)?
>
> **A3:** We sincerely thank you for highlighting the problem. Our work addresses **sample-level missing views**, where individual samples may lack specific view channels (e.g., a patient missing MRI data but having CT scans). Formally, given the indicator vector $r^{(i)}
> \in \mathbb{R}^{n_v}$ containing the index for $n_v$ existing samples for $v$-th view in sort, the indicator matrix $ \mathbf{G}^{(v)} \in \lbrace 0,1 \rbrace^{n \times n_v} $ satisfies:
> $$
> \mathbf{G}^{(v)}{i,j} = \begin{cases}
> 1, & \text{if the entry }  r^{(i)}{j}=i, \\\\
> 0, & \text{otherwise.}
> \end{cases}
> $$
>
> where $\mathbf{X}^{(v)}\mathbf{G}^{(v)}$ denotes the sorted complete data matrix in the $v$-th view.
>
> ---
>
> **Q4:** Table 2 shows empty results for FMVC on four datasets. Is this due to implementation errors or intentional omission?
>
> **A4:** We sincerely thank you for the question. The empty entries of FMVC in Table 2 stem from its prohibitive memory demands when handling large datasets. Specifically, FMVC constructs Laplacian matrices with size of  $n \times n$, which resulted in an out-of-memory error when running large-scale datasets. We will give further explanation in the final version.

---

> > ### Comment · Reviewer_9Soy · 2025-04-09
> >
> > Thank you for your detailed response—it has resolved my concerns. I would like to recommend the acceptance of this paper.

---

### Official Review · Reviewer_JeXy · 2025-03-19

**Overall Recommendation:** 3

**Summary:**

This paper proposes an Efficient Federated Incomplete Multi-View Clustering (EFIMVC) method to reduce communication overhead, enhance privacy protection, and handle missing views effectively. It employs a localized optimization strategy to lower communication costs, utilizes view-specific and shared anchor graphs to improve privacy, and introduces a dual-anchor alignment mechanism to enhance graph fusion stability. Experimental results demonstrate that EFIMVC outperforms existing methods in clustering accuracy, communication efficiency, and privacy preservation, highlighting its advantages in federated incomplete multi-view clustering tasks.

## update after rebuttal
Thanks for your careful response, and I consider the previous score reasonable and will keep the previous rating.

**Claims And Evidence:**

The main points of the paper are supported by convincing evidence. Extensive experiments on seven datasets demonstrate the superiority of the proposed method.

**Essential References Not Discussed:**

There are no related works that are not currently discussed in the paper.

**Experimental Designs Or Analyses:**

I checked the validity of the experimental designs and analyses. Extensive experiments are conducted on multiple widely-used multi-view clustering (MVC) benchmark datasets, with results averaged over multiple runs to ensure statistical reliability. The experiments cover different missing-view scenarios, ablation studies, and parameter sensitivity analysis to comprehensively evaluate the proposed method. The issues are listed behind in the Weaknesses.

**Methods And Evaluation Criteria:**

The proposed methods make sense for the problem. EFIMVC innovatively employs an anchor graph optimization strategy to reduce communication overhead and leverages local optimization to minimize data transmission, enhancing privacy protection. Additionally, the dual-anchor alignment mechanism ensures consistency between global and local anchors, thereby improving graph fusion quality. The experimental evaluation utilizes widely recognized multi-view clustering (MVC) benchmark datasets and state-of-the-art baseline methods to validate the effectiveness of the approach.

**Other Comments Or Suggestions:**

I would like to learn about the authors' response to the weaknesses listed above, which may give me a clearer perspective on the paper's contribution.

**Other Strengths And Weaknesses:**

This paper proposes an efficient federated incomplete multi-view clustering method (EFIMVC), which combines anchor graph optimization with a dual anchor alignment mechanism for the first time to reduce communication overhead, enhance privacy protection, and improve clustering performance in missing view scenarios. This method effectively reduces data transmission in federated learning while maintaining high clustering accuracy.
There are also some weaknesses：
1. It is recommended that all illustrations use vector graphics.
2. Some of the data sets listed are not the highest results. Can you add relevant explanations?
3. Although the dual anchor point alignment mechanism improves the quality of graph fusion, its computational complexity is not analyzed in detail. If there are more anchor points, the computational overhead may increase. It is recommended to provide theoretical analysis or experimental results.
4. The data distribution from different perspectives may be quite different, and the selection of anchor points may directly affect the final clustering effect. The paper does not discuss how to ensure the robustness of anchor point selection.

**Questions For Authors:**

I would like to learn about the authors' response to the weaknesses listed above, which may give me a clearer perspective on the paper's contribution.

**Relation To Broader Scientific Literature:**

The EFIMVC method first introduced anchor graph optimization and dual anchor alignment mechanism in the federated incomplete multi-view clustering problem. Based on the existing research on federated multi-view clustering, this method improves communication efficiency, privacy protection and missing view adaptability, filling the shortcomings of previous methods in these aspects.

**Theoretical Claims:**

I checked the correctness of the proofs for theoretical claims, including the effectiveness of the method of decoupling local optimization from global optimization in theoretical analysis, including the optimization problem solving process of quadratic programming.

---

> ### Author Rebuttal · Authors · 2025-04-01
>
> **We sincerely thank Reviewer JeXy for the thorough and constructive review. We provide point-by-point responses to the questions raised as follows:**
>
> ---
>
> **Q1:** It is recommended that all illustrations use vector graphics.
>
> **A1:** We sincerely appreciate your constructive suggestion. In the final version, we will convert all figures to vector graphics (e.g., PDF formats) to ensure optimal resolution and scalability.
>
> ---
>
> **Q2:** Some of the data sets listed are not the highest results. Can you add relevant explanations?
>
> **A2:** We sincerely thank you for raising this critical point. We clarify that the first four methods in Table 2 (FIMVC, SCBGL, DVSAI, DAQINT) are state-of-the-art centralized incomplete multi-view clustering methods. While they achieve marginally higher performance on specific datasets, they fundamentally address centralized scenarios and **cannot handle distributed multi-view data with privacy constraints**. In contrast, our federated framework achieves comparable or superior performance to these centralized SOTA methods while operating under strict distributed data storage protocols. This demonstrates our method’s unique capability to balance clustering accuracy with federated learning requirements. We will emphasize this distinction in the final version.
>
> ---
>
> **Q3:** Although the dual anchor point alignment mechanism improves the quality of graph fusion, its computational complexity is not analyzed in detail. If there are more anchor points, the computational overhead may increase. It is recommended to provide theoretical analysis or experimental results.
>
> **A3:** We sincerely appreciate the important technical concern. The computational complexity of the dual anchor alignment mechanism comes from optimizating the permutation matrix $\mathbf{P}^{(v)}$ on the server, which is $\mathcal{O}(m^3 + m^2n)$. Specifically, constructing matrix $\mathbf{Q}$  requires $\mathcal{O}(m^3 + m^2n)$ to process matrix multiplications. Solving $\mathbf{P}^{(v)}$ by performing SVD decomposition on $\mathbf{Q}$ requires $\mathcal{O}(m^3)$. While increasing anchor points $m$ raises time costs, our method fundamentally avoids the prohibitive space complexity ($\mathcal{O}(n^2)$) and time complexity ($\mathcal{O}(n^3)$) of full-graph approaches. We acknowledge the reviewer's concern and will explore hierarchical anchor optimization to further enhance efficiency while maintaining performance.
>
> ---
>
> **Q4:** The data distribution from different perspectives may be quite different, and the selection of anchor points may directly affect the final clustering effect. The paper does not discuss how to ensure the robustness of anchor point selection.
>
> **A4:** We sincerely thank you for the insightful comment. To evaluate anchor robustness, we conducted ablation studies using three strategies:
> 1. $k$-means (default)
> 2. Random selection
> 3. Density-based sampling
>
> The comparison results of different strategies are shown in below:
>
> |  **Datasets** | **ProteinFold** | **WebKB** | **100Leaves** |  **CCV**  | **Cifar10** |
> |:-------------:|:---------------:|:---------:|:-------------:|:---------:|:-----------:|
> |               |                 |    ACC    |               |           |             |
> |    $k$-means    |    **31.71**    | **90.64** |   **72.88**   | **20.04** |  **96.62**  |
> |     Random    |      30.83      |   76.48   |     69.27     |   15.53   |    96.18    |
> | Density-based |      30.99      |   79.63   |     69.63     |   15.53   |    96.13    |
> |               |                 |    NMI    |               |           |             |
> |    $k$-means    |    **40.33**    | **50.62** |   **84.76**   | **15.66** |  **91.52**  |
> |     Random    |       39.4      |   22.65   |     82.93     |   11.49   |    90.59    |
> | Density-based |       39.8      |   21.08   |     83.08     |   11.49   |    90.54    |
> |               |                 |   Purity  |               |           |             |
> |    $k$-means    |    **37.95**    | **90.64** |   **74.79**   | **23.45** |  **96.62**  |
> |     Random    |      36.87      |   81.81   |      71.3     |   19.22   |    96.18    |
> | Density-based |       37.1      |    80.5   |     71.61     |   19.36   |    96.13    |
> |               |                 |   Fscore  |               |           |             |
> |    $k$-means    |    **18.71**    |  **86.8** |   **61.84**   | **11.79** |  **93.46**  |
> |     Random    |        18       |   71.71   |     56.72     |    9.31   |    92.62    |
> | Density-based |      18.15      |   73.73   |     57.09     |    9.35   |    92.55    |
> | ||
>
> While $k$-means achieves the most stable performance, we acknowledge that advanced anchor initialization could further improve robustness. We will discuss this limitation and cite relevant techniques [1] in the final version.
>
> [1] Liu et al. Learn from View Correlation: An Anchor Enhancement Strategy for Multi-view Clustering. CVPR 2024.

---

### Decision · Program_Chairs · 2025-05-01

**Decision:**

Accept (poster)

**Comment:**

After the rebuttal, all reviewers agreed to accept this paper. To overcome the problems of excessive communication overhead, insufficient privacy protection, and inadequate handling of missing views, this paper proposes Efficient Federated Incomplete Multi-View Clustering (EFIMVC) that introduces a localized optimization strategy to significantly reduce communication costs while ensuring theoretical convergence. The paper is well-written. The incomplete view scenarios are valuable.